# Interrelationship between COVID-19 and Coagulopathy: Pathophysiological and Clinical Evidence

**DOI:** 10.3390/ijms24108945

**Published:** 2023-05-18

**Authors:** Beatrice Ragnoli, Beatrice Da Re, Alessandra Galantino, Stefano Kette, Andrea Salotti, Mario Malerba

**Affiliations:** 1Respiratory Unit, Sant’Andrea Hospital, 13100 Vercelli, Italy; beatricedare95@gmail.com (B.D.R.); alessandragalantino@gmail.com (A.G.); stefano.kette@libero.it (S.K.); salottiandrea@gmail.com (A.S.); mario.malerba@uniupo.it (M.M.); 2Department of Traslational Medicine, University of Eastern Piedmont (UPO), 28100 Novara, Italy

**Keywords:** COVID-19 infection, coagulopathy, endothelial dysfunction, platelet activation, citokine storm, anticoagulant therapy

## Abstract

Since the first description of COVID-19 infection, among clinical manifestations of the disease, including fever, dyspnea, cough, and fatigue, it was observed a high incidence of thromboembolic events potentially evolving towards acute respiratory distress syndrome (ARDS) and COVID-19-associated-coagulopathy (CAC). The hypercoagulation state is based on an interaction between thrombosis and inflammation. The so-called CAC represents a key aspect in the genesis of organ damage from SARS-CoV-2. The prothrombotic status of COVID-19 can be explained by the increase in coagulation levels of D-dimer, lymphocytes, fibrinogen, interleukin 6 (IL-6), and prothrombin time. Several mechanisms have been hypothesized to explain this hypercoagulable process such as inflammatory cytokine storm, platelet activation, endothelial dysfunction, and stasis for a long time. The purpose of this narrative review is to provide an overview of the current knowledge on the pathogenic mechanisms of coagulopathy that may characterize COVID-19 infection and inform on new areas of research. New vascular therapeutic strategies are also reviewed.

## 1. Background

At the end of December 2019, a novel coronavirus, denominated SARS-CoV-2 according to the similarity with the previous SARS viral epidemy, was described for the first time in China, and in March 2020, it was declared a global pandemic by the WORLD Health Organization (WHO) due to high morbidity and mortality. Italy was one of the most affected countries at the beginning of the infection spreading [1]. The disease caused by this virus was denominated Coronavirus disease 2019 (COVID-19) which still represents a critical challenge for the worldwide health community despite the reduction in mortality following the global vaccination campaign. To date, there have been more than 762 million confirmed cases of COVID-19 infection, including almost 6.8 million deaths reported to WHO, while a total of 13,340,275,493 vaccine doses have been administered [2]. Among the wide range of SARS-CoV-2 clinical manifestations (cough, fever, pharyngodynia, myo-arthralgia, fatigue), a respiratory tract involvement has been observed, potentially evolving with pneumonia, acute respiratory distress syndrome (ARDS), hyperinflammation, and COVID-19-associated-coagulopathy [3,4,5]. The hypercoagulable state is strongly associated with COVID-19 infection and may explain several phenomena observed in clinical practice. Since the beginning of the pandemic, a very high incidence of thrombo-embolic events was observed including arterial and venous thrombosis, cerebral and myocardial infarction, limb arterial thrombosis, and venous thrombosis leading to a higher incidence of stroke, acute coronary syndrome and myocardial infarction, venous thromboembolism (VTE) and pulmonary thromboembolism (PTE) [6,7,8,9,10,11]. Pathophysiological characteristics of acute pulmonary thromboembolism [12] and abnormal coagulation status [13] have been reported in these patients. Additional important laboratory findings such as high levels of D-dimer, as well as fibrinogen (FIB) and its related degradation products (FDP), have been correlated with a poorer outcome [13,14,15]. In patients who died from COVID-19 infection, microthrombosis of alveolar capillaries was more prevalent (nine times) than in patients who died from influenza, and about 15.2% to 79% of patients with severe COVID-19 infection have shown thrombotic events [16]. The involvement of the coagulation cascade and its abnormalities were previously identified in experimental investigations on mice infected with SARS-CoV-1 and in human autopsies. These findings suggested the hypothesis of a diffuse thrombotic microangiopathic mechanism involved in the pathogenesis of acute pulmonary interstitial disease caused by SARS-CoV-2 infection [17,18]. The prothrombotic status seems to be caused by immune cell activation, excessive coagulation, and endothelial dysfunction [19]. Immuno-thrombosis appears to be involved in the pathological mechanism of SARS-CoV-2, and it is characterized by the interaction between the hemostatic system and the innate immune system, especially between monocytes, macrophages, and neutrophils. After the endocytosis of SARS-CoV-2 in the host cells, vascular damage is induced, leading to a proinflammatory form of programmed cell death with cell lysis named “pyroptosis” and the release of various substances, the so-called damage-associated molecular patterns (DAMPs) such as adenosine triphosphate (ATP), nucleic acids, and inflammasomes [20], thus intensifying the inflammatory environment. Several mechanisms have been hypothesized to be involved in this hypercoagulable process such as inflammatory cytokine storm, platelet activation, endothelial dysfunction, and stasis for a long time [11,21,22]. The purpose of this narrative review is to provide an overview of the current knowledge on the pathogenic mechanisms of coagulopathy that may characterize COVID-19 infection and inform on new areas of research. New vascular therapeutic strategies are also reviewed.

## 2. Role of Platelets and Complement as Prothrombotic Factors in COVID-19 Infection

Platelets have a pivotal role in the innate immune system by activating the complement, thus playing a key role in COVID-19 “immune-thrombosis” [23]. The aggregation of PLT activated by endothelium damage and its interaction with other cells increase their potential for pathologic thrombosis; their activation is essential to the structural remodeling of the pulmonary vasculature, inflammation, and cardiovascular disease [24,25]. The mechanism of platelet activation may include different and multiple pathways even more complex in COVID-19 infection, as the virus is able to infect cells using several entry mechanisms such as TLRs and/or the ACE2-AngII axis [26]. The activated endothelial cells express P-selectin and other adhesion molecules with the recruitment of platelets and leukocytes. Bioactive molecules (e.g., adenosine diphosphate [ADP], polyphosphates, coagulation factors) and immunological mediators (e.g., complement factors) are released from activated platelets, activating the immune system through positive feedback [23]. P-selectin, a platelet activation marker, is increased in patients with COVID-19 and can lead to a procoagulant phenotype by inducing tissue factor (TF) expression in monocytes. Moreover, Von Willebrand factor (VWF) is a glycoprotein derived from activated endothelial cells, platelets, or sub-endothelial cells mediating the adhesion and aggregation of platelets. In patients affected by COVID-19, VWF is significantly increased and may suggest a tendency for thrombosis [27]. The activation of the complement system was documented in COVID-19 with the formation of the terminal membrane attack complex (MAC) that, in turn, can activate platelets with subsequent endothelial damage and the secretion of VWF [28]. For this reason, the complement activation is associated with an amplification of the prothrombotic phenotype in COVID-19. In fact, C5a can stimulate the release of TF and plasminogen activator inhibitor-1 (PAI-1) expression and activate neutrophils, which are responsible for the increased release of cytokines and the formation of neutrophil extracellular traps (NETs) [29].

## 3. Role of Hypoxia, Blood Viscosity, and Vasoconstriction as Prothrombotic Factors in COVID-19 Infection

Hypoxia may represent itself as a factor inducing a prothrombotic status in patients with SARS-CoV-2 infection with the production of a hypoxia-inducible transcription factor (HIF-1α), which promotes the secretion of PAI-1 (plasminogen activator inhibitor) and macrophages by the endothelium. On the other hand, the mechanisms of altered coagulation are responsible for hypoxia that, in turn, favors the thrombo-inflammatory loop. Furthermore, hypoxia causes a release of cytokines such as tumor necrosis factor-α (TNF-α) IL-6 [30], critical inflammatory cytokines with prothrombotic effects. Positive correlations have been found between IL-6 and D-dimer, especially during the exacerbation of the disease [31]. Consequently, increased blood viscosity and the release of procoagulant antibodies develop [32]. Recent studies showed that the appearance of antiphospholipid antibodies and lupus anticoagulant immunoglobulins may also play a role in the pathogenesis of coagulopathy. Indeed, the presence of IgA anti-cardiolipin antibodies and IgA and IgG anti-2-glycoprotein I antibodies have been found in association with coagulopathy, thrombocytopenia, and the development of peripheral and cerebral ischemic events. Harzallah and coworkers [33] investigated 56 patients with confirmed or suspected SARS-CoV-2 infection. Among these, 25 were found to be lupus anticoagulant immunoglobulin, while 5 were found positive for IgM or IgG anti-cardiolipin or anti-2-glycoprotein I antibodies. Endothelial dysfunction is characterized by the loss of characteristics of endothelial native cells such as the ability to regulate vascular tonus which may conduce to vasoconstriction and, subsequently, a prothrombotic status. Moreover, the down-regulation of the endothelial ACE2 receptor as a consequence of SARS-CoV-2 infection gives a pro-inflammatory, pro-coagulant, and pro-apoptotic phenotype to endothelial cells [34].

## 4. Interlink between Coagulation and Inflammation in COVID-19 

The so-called COVID-19-associated CAC represents a key aspect in the genesis of organ damage from SARS-CoV-2 and the hypercoagulation state is based on an interaction between thrombosis and inflammation. A close relationship between inflammation and coagulation has been widely demonstrated in previous research [35,36]. The coagulation system consists of a finely regulated balance between procoagulant and anticoagulant mechanisms and inflammation can compromise this equilibrium, leading to impaired coagulation. As a result, the final clinical consequence of inflammatory conditions may consist of bleeding, thrombosis, or both of them [37]. Pathogens, inflammatory mediators such as IL-6, IL-8, and TNF-α, as well as DAMPs from injured host tissue can activate monocytes and induce the expression of tissue factors on monocytes and endothelial cell surfaces [38] (Figure 1).

Subsequently, activated monocytes release inflammatory cytokines and chemokines that enlarge the inflammatory response and stimulate neutrophils, lymphocytes, platelets, and vascular endothelial cells. Healthy endothelial cells have an anti-thrombogenic attitude due to the expression of glycocalyx and its binding protein, antithrombin. When endothelial cells go through injury, the glycocalyx is disrupted, the anticoagulant factors are lost, and, consequently, these cells change their properties to procoagulant [39]. Furthermore, neutrophils are also involved in an important defense mechanism that may lead to a procoagulant status by means of NETs. NETs are structures of DNA, histones, and neutrophil antimicrobial proteins that bind and kill pathogens. The excessive production of NETs can facilitate microthrombosis by creating a scaffold for platelet aggregation [40]. When an infection occurs, the first leukocytes recruited are neutrophils that, producing and releasing NETs, stimulate the formation and deposition of fibrin to trap and destroy invading microorganisms. It has been previously demonstrated that NETs increase in sepsis and inflammatory conditions [41]. NETs also cause platelet adhesion, and, in some experimental models, their connection with deep vein thrombosis has been demonstrated [40]. They stimulate both the extrinsic and intrinsic coagulation pathways playing a major role in a coagulative pattern during infection-mediated inflammation. Patients with severe COVID-19 have been shown to present elevated levels of circulating histones and myeloperoxidase DNA (MPO-DNA) which are two specific markers of NETs [42]. As a consequence of the described mechanisms, an extreme inflammatory response may also occur, causing disseminated intravascular coagulation (DIC), which leads to multiple organ failure. This life-threatening acquired syndrome is characterized by the disseminated and often uncontrolled activation of coagulation and is associated with a high risk of macro- and microvascular thrombosis. In this setting, natural coagulation inhibitors also become inefficient in downregulating thrombin generation. Moreover, progressive consumption coagulopathy can be observed which leads to an increased bleeding risk [43]. Other clinical manifestations of the altered coagulation system are hemolytic uremic syndrome, idiopathic thrombocytopenic purpura, thrombotic thrombocytopenic purpura [44], and hemophagocytic lymphohistiocytosis (sHLH). Globally, all of this evidence suggests that the hypercoagulative state described in patients with COVID-19 is likely to be caused by a deep and complex inflammatory response to the virus, based on an interaction between thrombosis and inflammation as shown in Figure 2. Another important interlink between inflammation and pro-thrombotic status is represented by underlying clinical conditions such as chronic comorbidities that are linked to mortality in COVID-19 infection. In particular, obesity has been shown to increase the risk of hospitalization and COVID-19 complications [45] suggesting an interplay between obesity and inflammation. The adipose tissue, in fact, expresses higher ACE2 levels than lung tissue, being a powerful inflammatory reservoir for the replication of SARS-CoV-2 [46]. In addition, obese people are characterized by low-grade inflammation, associated with the over-expression of pro-inflammatory cytokines and chemokines such as TNF-α, IL-6, and MCP-1, high leptin levels with known pro-inflammatory effects, low adiponectin levels with anti-inflammatory effects, and, consequently, a procoagulant status. It has been calculated that one-third of total circulating concentrations of IL-6 originate from adipose tissue [47]. In addition, obese patients show higher blood IL-6 and TNF-α levels and a polarization of natural killer (NK) cells to non-cytotoxic NK cells. As both obesity and COVID-19 seem to share common metabolic and inflammatory pathways, it has been recommended by many authors to consider and classify obese and severely obese patients as high-risk patients for COVID-19. Additionally, sleep disturbances during pandemics have been suggested to be related to a major risk of infection linked to increased inflammatory status and a reduction in the efficiency of the immune system [48]. An interesting linkage was found between sleep deprivation, inflammation, and immune response to SARS-CoV-2 that may have a role in predisposing to the infection [49].

## 5. Interlink between Coagulopathy in Viral Infections and in COVID-19 

Since the beginning of the pandemic, a very high incidence of thrombo-embolic events (VTE) was observed. The hypercoagulative state, described in patients with COVID-19 derives from a complex inflammatory response to the virus in which hemostasis and the immune system collaborate together to limit the spread of viral infection. Physiological immune thrombosis can evolve into an excessive, dysregulated formation of immunologically mediated thrombi and spread, especially in the microcirculation. Several viral infections may share abnormal coagulation processes such as bleeding, thrombosis, or both.

### 5.1. Thrombosis

The increased incidence of VTE in COVID-19 patients was similar also in patients with other viral infections, i.e., severe acute respiratory syndrome (SARS) and Middle East Respiratory Syndrome (MERS-CoV) [50,51]. H1N1 influenza infection is associated with an 18-fold increased risk of developing VTE when compared to critically ill patients with ARDS with no H1N1 influenza infection [6]. A previous study by Avnon et al. found that VTE occurred in 25% of patients with severe H1N1 influenza admitted to the intensive care unit (ICU) [52]. Particular evidence for thromboembolic events was also reported during cytomegalovirus (CMV) infection in which two arterial thrombotic events were described in nine Israelitic immunocompetent CMV-infected patients (spleen and liver) [53]. The pathophysiological mechanism is yet unknown but it seems to be related to higher levels of VWF in the plasma of CMV-infected people [54]. It is likely that the SARS-CoV-2 virus does not have intrinsic procoagulant effects, while coagulopathy appears as a consequence of the intense COVID-19 inflammatory response and endothelial activation/damage [55]. Two possible mechanisms implicated in the pathogenesis of coagulation dysfunction during SARS-CoV2 infection have been proposed: the cytokine storm which seems to play a pivotal role, and virus-specific mechanisms related to the virus interaction with the renin–angiotensin system and the fibrinolytic pathway [56].

#### 5.1.1. Cytokine Storm

Pro-inflammatory cytokines are involved in a so-called “cytokine release syndrome” responsible for the innate immune system activation and severe clinical manifestation of the disease [57]. Immune system dysfunction is a candidate risk factor for adverse outcomes in COVID-19, and the most important cause of morbidity and mortality in patients suffering from COVID-19 infection seems to be the cytokine storm causing an immune dysregulation in the peripheral tissues and in the lungs [58] (p. 2), refs. [57,59,60,61,62]. More specifically, IL-6 plays an important role in cytokine release syndrome and contributes, together with TNF-α and interleukin-1 (IL-1), to blood hyper-coagulability and to severe inflammation, sometimes evolving in disseminated intravascular coagulation (DIC) [63] Figure 3.

Current evidence from clinical studies shows that IL-6 seems to play a prominent role in the cytokine-induced activation of coagulation. Additionally, IL-6 promotes the proliferation of megakaryocytes [64] and the release of TF, the latter detected in inflamed tissues and in particular in the lungs of patients affected by COVID-19 [65]. A postulated mechanism considers that SARS-CoV-2-infected megakaryocytes may interfere with platelet function and count, as already described in previous studies that reported thrombocytopenia during SARS-CoV infection. The virus induces the release of cytokines such as IL-6 conducting to megakaryocytic proliferation and differentiation, although the mechanism remains not completely clarified [66,67].

Furthermore, vascular permeability is mediated by IL-6 through the stimulation of vascular endothelial growth factor (VEGF) secretion and the release of other coagulation factors such as FIB and factor VIII [68]. There was a great effort during the pandemic to find inflammatory markers reflecting disease severity and eventually predicting disease prognosis. Among the most studied, increased levels of a pivotal serum cytokine, IL-1, which is a principal source of tissue damage interacting in both innate and acquired immunity, have been detected in patients suffering from severe COVID-19 infection. IL-1 stimulates the secretion of mediators stored in the granules of mast cells and macrophages, such as TNF-α, IL-6, and the release of arachidonic acid products such as prostaglandins and thromboxane A2 [69,70,71,72]. Another important marker in the cytokine network of COVID-19 infection is IL-18. The catastrophic clinical course of COVID-19 shares similar features with macrophage activation syndrome (MAS) encountered also in other conditions with a potentially rapidly fatal course without treatment. IL-1, IL-6, IL-8, IL-10, IL-18, interferon (IFN)-γ, and TNF-α are the most important elements responsible for MAS development. IL-18 is produced by macrophages at very early stages of viral infections and induces the production of IL-6 and IFN-γ which are considered critical for optimal viral host defense. A study by Satis and coworkers observed a four-fold level of IL-18 in 58 people suffering from a severe form of COVID-19. These findings contrasted with the mildly affected patients and led to the conclusion of a correlation between IL-18 and the severity of the disease [73]. An additional role is determined by TNF-α, responsible for the activation of glucuronidases, which degrades the endothelial glycocalyx, and the upregulation of hyaluronic acid synthase 2, which leads to hyaluronic acid deposition and fluid retention [74]. Due to the systemic hypoxia induced by COVID-19-related ARDS, a reduction in endothelial nitric oxide synthase activity and nitric oxide levels has been indicated as a possible pathogenic process typical of endothelial dysfunction [75].

#### 5.1.2. Virus-Specific Mechanisms

Experiments in vitro demonstrated that SARS-CoV2 can infect primary endothelial cells [76] and there is some evidence of the infection of endothelial cells in severe cases of COVID-19 [11]. Moreover, the replication within endothelial cells is able to induce cell death causing the activation of procoagulant reactions [77]. The membrane glycoprotein (Spike) of the SARS-CoV-2 virus interacts with Angiotensin-Converting Enzyme 2 (ACE-2), an integral membrane receptor expressed in the lung but also the heart, kidney, and intestine by reducing their activity. Normally, ACE-2 reduces the availability of angiotensin II through the counter-regulated activity of ACE [78]. As a result, the virus-mediated engagement of ACE-2 decreases its expression and activates the renin-angiotensin system (RAS), promoting the activation of epithelial cells, monocytes, neutrophils, and procoagulant factors with platelet adhesion and aggregation, and consequent vasoconstriction and release of inflammatory cytokines [79], as well as a reduction in fibrinolytic activity mediated by RAS, can be observed [80] as represented in Figure 4.

The RAS may play a key role in SARS-CoV-2-induced COVID-19 [81]. The downregulation of ACE-2 by the virus causes an increase in angiotensin II, which, acting on the AT1 receptor, causes systemic injury [82] but also specific lung damage with pulmonary fibrosis, pulmonary inflammation, and ARDS in severe cases of COVID-19 [83]. ACE-2 is markedly expressed in pneumocytes type II, hence participating in alveolar surfactant production. The downregulation of ACE-2 receptors due to the binding of coronavirus might hinder the expression of pneumocytes type II cells, explaining the worsening of gaseous exchange [84,85]. Overall, the interaction of coronavirus with ACE-2 receptors is destructive due to increased inflammatory lesions, the downregulation of ACE-2 receptors, increased local angiotensin II effects and AT1 receptor over-activity, insufficient surfactant due to bruised pneumocytes type II causing a reduction in pulmonary compliance and amplified surface tension, and a reduction in the generation and repair of pneumocytes type I with impaired gaseous exchange along with alveolar–capillary diffusion capacity and fibrosis [86]. Moreover, a different impact of SARS-CoV-2 expression on ACE-2 may be due to gender-related dissimilarities, with the ACE-2 gene existing in the X-chromosome [87]. The wide variances in COVID-19 death rates might be explained by significant alterations in the equilibrium of the ACE:ACE-2 system associated with gender, racial, and age differences in genetic ACE and ACE-2 polymorphism and environmental aspects manipulating ACE-2 expression [88,89,90]. In addition, the severity of lung injury is linked with the expression of ACE. ALI was less complicated in complete knockout (Acee/e) mice and AT1 receptor knockout mice compared to partial ACE knockout (Ace./e) mice and wild-type mice, respectively. The injection of recombinant SARS spike protein along with AT1 blockers elevated the expression of angiotensin II leading to ARDS in mice [91]. Thus, understanding the role of the ACE-2 receptor in the pathogenesis of COVID-19 may open a potential approach for therapeutic intervention [92].

Among virus-related mechanisms, high levels of PAI-1, the principal inhibitor of fibrinolysis interfering with tissue plasminogen activator (tPA) and urokinase, have been related to an increased risk of thromboembolic events [80]. Interestingly, previous studies reported high blood levels of PAI-1 in patients with SARS-CoV infection suggesting a possible direct effect of infection on the production of anti-coagulant factors [93]. One study described an important increase in another mediator of platelet adhesion, platelet-derived vitronectin (VN), in SARS-CoV pneumonia; however, it was not possible to discriminate its origin from increased expression by the liver or from lung damage [94]. Another possible virus-specific effect could be related to the induction of autoimmunity, also described in SARS patients [37]. Recent studies showed that the appearance of antiphospholipid antibodies and lupus anticoagulant immunoglobulins may have a role in the pathogenesis of coagulopathy. Indeed, the presence of IgA anti-cardiolipin antibodies and IgA and IgG anti-2-glycoprotein I antibodies have been found in association with coagulopathy, thrombocytopenia, and the development of peripheral and cerebral ischemic events. Antiphospholipid antibodies (aPL), recognized as risk factors for arterial and venous thrombosis, have been associated with different viral infections, such as parvovirus B19, herpes viruses, hepatitis viruses, and human immunodeficiency viruses. The first case report of a COVID-19 patient with aPL and arterial ischemia was described by Chinese authors [95], although, subsequently, a larger, multicentric cohort demonstrated a low rate of aPL positivity, as defined by classification criteria, suggesting that aPL found in COVID-19 patients is different from aPL found in antiphospholipid syndrome [96]. It is likely that the mechanisms of altered coagulation due to SARS-CoV-2 infection, also responsible for hypoxia, may in turn favor the thrombo-inflammatory loop and consequently increased blood viscosity and the release of procoagulant antibodies [32]. These observations were confirmed by a study by Harzallah and coworkers investigating 56 patients with confirmed or suspected SARS-CoV-2 infection. Among these, 25 were found with lupus anticoagulant immunoglobulin, whereas 5 were found positive for IgM or IgG anti-cardiolipin or anti-2-glycoprotein I antibodies [33]. Further studies are needed to address this issue.

### 5.2. Thrombocytopenia

COVID-19-related coagulopathy firstly determines elevated D-dimer levels that combine in turn with mildly prolonged PT, APTT, and mild thrombocytopenia. At late stages, this process evolves into a classical DIC [97]. These findings were identified in the clinical setting in a meta-analysis where 7.613 patients suffering from COVID-19 infection were examined. In this cohort, thrombocytopenia was worse in the critically ill group than in those with non-severe disease [98]. Additionally, the platelet count was lower in the elderly, in males, and in patients with higher APACHE II scores at admission [99]. This study highlights an association between low platelet counts and an increased risk of severity of the disease and mortality. As per SARS-CoV-2-infection-related thrombocytopenia, it appears that the platelets can be more rapidly removed or sequestrated by the reticuloendothelial system after the activation of antigen–antibody complexes [100,101]. Additionally, the megakaryocyte’s function and the consequent platelet production can be reduced by the virus activity [102]. A possible mechanism of thrombocytopenia was described after COVID-19 vaccination. It was observed in rare cases that immune thrombotic thrombocytopenia (VITT) syndrome was induced by the vaccine, particularly the ChAdOx1 nCoV-19 vaccine. The main pathogenetic hypothesis supporting this evidence is the possible promotion of antibody synthesis against PF4 by some anti-COVID vaccines promoting the synthesis of antibodies against PF4 that provoke platelets’ massive activation, inducing immune thrombotic thrombocytopenia [103]. As anti-PF4 antibodies were detected in patients with VITT, the current guidelines recommend a PF4-heparin ELISA blood test before performing a vaccine when VITT is clinically suspected [104]. The risk of clotting in the general population is estimated to be around 1:250,000, although it is higher in young people (20–29 years old) at 1.1:100,000 [105].

## 6. Contribution of Sepsis in Coagulopathy during COVID-19 Infection

Sepsis is a life-threatening condition as a response to a primary infection in which the body responds with extreme inflammatory reactions that create injuries in one’s own tissues and organs. On the other hand, severe COVID-19 infection is commonly complicated with coagulopathy, and, in the latter stages, may evolve towards a classical DIC. These manifestations were an object of major concern during the COVID-19 pandemic. The International Society of Thrombosis and Hemostasis (ISTH) has proposed a new category to identify an early stage of DIC associated with sepsis called sepsis-induced coagulopathy (SIC). Many patients suffering from severe COVID-19 meet the Third International Consensus Definitions for Sepsis (Sepsis-3) [106] manifesting respiratory dysfunction during a viral infection. The diagnostic criteria of SIC are summarized in Table 1. A score ≥ 4 is diagnostic for SIC. This score can also be applied to COVID-19-affected patients to identify a coagulopathy risk induced by the virus, although it is less reliable than in other pathogen-induced infections as, in this case, especially during the initial stages of the disease, thrombocytopenia cannot be present. One study by Tang et al. studied the effects of anticoagulant treatment to validate the usefulness [107,108] of the SIC score, finding that patients who met the criteria reported in Table 1 benefit from anticoagulant therapy [12].

### Coagulation Biomarkers in SARS-CoV-2 Infection: A Predictive Method

In the setting of the altered coagulation state, the measurements of the coagulative parameters may orient the clinicians toward the early identification of a coagulative derangement. Besides the D-dimer, as above mentioned, other parameters are of bedside interest (Table 2). Increased levels of thrombin–antithrombin complexes, plasmin-alpha-2-antiplasmin, and thrombomodulin complexes have been reported in respiratory tract infections. Increased PAI-1 serum levels were identified, suggesting impaired fibrinolysis. A study [15] highlighted an alteration of the laboratory parameters deponent for DIC (according to the diagnostic criteria of the ISTH) in 15 subjects (71.4%) who died of COVID-19-related pneumopathy. In the final stage of the disease, elevated levels of D-dimer and FIB degradation products were found. Recent contributions have reported that COVID-19 severity could be associated with some coagulopathy biomarkers, including prothrombin time (PT), activated partial thromboplastin time (APTT), and D-dimer. Nevertheless, the association between coagulopathy and COVID-19 severity still remains undefined.

The severity of the condition is mostly associated with clinical evidence (Table 3). In particular, one study [109] demonstrated that the majority of patients developed a mild infection, and about 15% of them experienced a severe manifestation with dyspnea and hypoxia. Another 5% developed respiratory failure in conjunction with ARDS, shock, and multi-organ dysfunction. Many studies have focused on the evaluation of D-dimer, PLT, PT, APTT, and FIB. It was reported that D-dimer and PT values have been shown to be higher in patients with more severe disease [110]; moreover, several studies have shown that elevated D-dimer levels are associated with in-hospital mortality. Recent research studies have hypothesized that genetic profiles may partly explain individual differences in developing thrombotic complications during COVID-19 infection. An interesting study evaluated the genotypic distribution of targeted DNA polymorphisms in COVID-19 complicated by pulmonary embolism during hospitalization, finding significant associations between higher D-dimer levels and ACE I/D and APOE T158C polymorphism in patients with and without pulmonary embolism, suggesting a potentially useful marker of poor clinical outcomes [111]. Previous data showed a higher prevalence of ACE D/D genotype in severe COVID-19 patients compared to those with mild disease; this genotype is significantly associated with cardiometabolic diseases and obesity, known risk factors for COVID-19 [112,113,114,115]. Additionally, this genotype was associated with thrombo-embolic manifestations in patients affected by other diseases and traditional thrombophilia-related polymorphisms [116], increased venous thromboembolism risk [117,118], and endothelial damage with hypercoagulability in patients with arterial hypertension [119]. The APOE locus has been associated with increased vulnerability to severe COVID-19 mortality, especially for the APOE4 homozygous genotype [120] which is the strongest genetic risk factor for sporadic Alzheimer’s disease. This appears to be very important from a clinical point of view as recent data show that dementia can predict the severity of COVID-19 infection. In fact, patients with dementia are more exposed to the severe form of the infection and are more likely to require hospitalization and to have severe sequelae or fatal outcomes compared with patients who do not [5,121]. Finally, the racial variance of ACE I/D genotype polymorphism seems to be correlated with different outcomes during COVID-19 infection; in fact, populations with higher D allele frequency (e.g., Italian) experienced higher fatality [122]. In another meta-analysis, it was demonstrated that the platelet count decreased progressively with the degree of disease severity [123]. However, a previous meta-analysis [124] demonstrated that there were no differences in PLT and APTT levels between wild and severe cases. All this is probably due to the confounding factors and biases that inevitably occur, such as age, sex, and the presence of comorbidities such as hypertension, diabetes, cardiovascular disease, and chronic kidney disease of the examined populations. As reported in another study by Wu et al., mortality from severe COVID-19 was increased 34-fold compared to a normal infection [125] and very high levels of coagulation markers were correlated with an 11-fold increase in death. These observations underline the importance of the early stratification of disease severity.

## 7. New Clinical Evidence of Anticoagulant Therapy in COVID-19

Data on anticoagulant therapy appear to be associated with a better outcome in moderate-to-severe COVID-19 patients with altered coagulative parameters (elevated D-dimer, elevated FIB, and low levels of anti-thrombin) [13,14,132]. A retrospective study by Shi et al. showed that these treatments can mitigate cytokine storm exerting an anti-inflammatory effect (reduction in IL-6 and increase in lymphocytes) and improving coagulation dysfunction [133]. A number of substances are used for COVID-19 VTE such as heparins, direct oral anticoagulants (DOAK), aggregation inhibitors, factor XII inhibitors, thrombolytic agents, anti-complement, anti-NET drugs, and IL-1 receptor antagonists.

Heparins, including unfractionated heparin (UFH) and low-molecular-weight heparin (LMWH), have several anti-coagulant and anti-inflammatory effects [134]. Among the various properties of heparin, a beneficial effect on endothelium has been observed. Dysfunctional endothelium leads to an inflammatory status through the production of vasoconstrictor factors and the recruitment of immune cells [135]. Histones released from damaged cells may be responsible for endothelial injury [136]. Heparin exerts its action through an effect on histone methylation and MAPK and NF-κB signaling pathways [137]. In this way, heparin can antagonize histones and therefore “protect” the endothelium [29,30]. It was proved to have a beneficial effect related to its anticoagulant function on COVID-19 [138] and anti-inflammatory properties [139]. The proposed mechanisms include binding to inflammatory cytokines, the inhibition of neutrophil chemotaxis and leukocyte migration, the neutralization of complement factor C5a, the sequestration of acute-phase proteins such as P-selectin and L-selectin, and the induction of cell apoptosis through the TNF-α and NF-κB pathways [140,141]. Another potential direct antiviral role of heparin is related to its polyanionic properties allowing it to bind to various proteins thus acting as an effective inhibitor of viral adhesion [142]. This condition mechanism was also described in other viral diseases [142,143] as well as in SARS-CoV. As Mycroft-West et al. [144] demonstrated, surface plasmon resonance and circular dichroism were used, and it was demonstrated that the receptor binding domain of the Spike S1 SARS-CoV-2 protein interacts with heparin. In a report by Tang [15], a favorable outcome was highlighted with the use of LMWHs in severe patients with COVID-19 who meet the criteria of SCI (sepsis-induced coagulopathy) or with markedly elevated D-dimer. A large, retrospective multicentric study among in-hospital patients (the CORIST study) showed that heparin treatment was associated with lower mortality, particularly in severely ill COVID-19 patients and in those with strong coagulation activation [145]. Moreover, research conducted in the neurorehabilitation department of a neuroscience referral hospital following neurological damage showed, despite a small number of patients, that hospitalized, vulnerable, patients with severe neurological damage can present a completely unexpected benign disease course of SARS-CoV-2 infection after heparin treatment. The anti-inflammatory and anticoagulant effects of enoxaparin administered much earlier before and during the infection, together with possible antiviral activity, could explain the favorable disease course observed in severe neurological patients with an increased risk of poor outcomes. Further research is needed to explore the possible mechanisms of action of enoxaparin in critical neurological patients with COVID-19 and confirm these observations [146].

However, several studies could not identify this relationship. As demonstrated by C. Coligher et al. in a randomized control trial, in critically ill patients with COVID-19, an initial strategy of therapeutic-dose anticoagulation with heparin failed to show a greater probability of survival to hospital discharge or a major number of days free of cardiovascular or respiratory organ support than usual-care pharmacologic thromboprophylaxis [147]. Interim results from multiplatform RCTs on VTE prophylaxis show that in moderate COVID-19 (hospitalized, not intensive), therapeutic doses of LMWH appear to be better than prophylactic doses, with positive effects on morbidity and mortality and less than 2% severe bleeding [148]. In patients at low or intermediate risk of thrombotic phenomena, treatment with prophylactic doses of LMWH has been noticed to produce a concomitant reduction in developing severe ARDS and venous thromboembolism, which may reduce the need for mechanical ventilation and consequentially lower cardiovascular death [149]. Treatment with heparin did not improve the course of severe COVID-19 and it seems to be inferior to prophylactic doses. The first observational cohort study examined previous prophylactic anticoagulation versus no anticoagulation in hospitalized COVID-19 patients (not intensive). Early treatment with prophylactic heparin was associated with a 34% reduction in relative 30-day mortality risk and an absolute risk reduction of 4.4%. There was no increased risk of bleeding under prophylactic anticoagulation [150]. Guidelines of medical societies currently recommend VTE prophylaxis, preferably with LMWH, for every inpatient COVID-19 patient [151]. The guidelines do not recommend VTE prophylaxis for COVID-19 outpatients. Prophylactic anticoagulation for 1–2 weeks is recommended by some guidelines in patients discharged from hospitals if there are additional risk factors [152]. Globally, the use of heparin is recommended, but it needs to be titrated against the risk of bleeding and individualized, especially in patients affected by pre-existing endothelial dysfunction (diabetes, hypertension, obesity) at higher risk of adverse outcomes during COVID-19 infection [153]. Additionally, antiplatelets have been considered an antithrombotic treatment for COVID-19, even though the rationale for aspirin use in COVID-19 is still uncertain. A recent review [154] recommends a low-dose aspirin regimen for the primary prevention of arterial thromboembolism in patients aged 40–70 with intermediate or high atherosclerotic cardiovascular risk and a low risk of bleeding. This opens a perspective on aspirin’s protective role in COVID-19 with associated lung injury and vascular thrombosis even in the absence of previously known cardiovascular disease.

The contact activation system, including factor XII (FXII), factor XI (FXI), high-molecular-weight kininogen, and prekallikrein, links inflammation and coagulation, triggering thrombin generation which promotes platelet activation but also upregulates the kallikrein–kinin system (KKS) which induces the renin–angiotensin system with the release of pro-inflammatory cytokines [155]. The inhibition of contact activation has been shown, especially in animal models, to prevent consumptive coagulopathy, pathologic systemic inflammatory response, and mortality [156]. Direct FXa inhibitors have been already shown to possess an inflammatory and antiviral effect in addition to their well-established anticoagulant activity, and they have been proposed to have a potential therapeutic role in coronavirus infections [157]. FXI activation by virtue of its position as an interface between contact activation and thrombin generation has been suggested as a unique and promising target to safely prevent or treat COVID-19-related inflammatory complications including cytokine response and coagulopathy, hence reducing associated mortality, and, evidence from recent research suggests that the inhibition of FXIa seems to attenuate thrombosis with little effect on hemostasis and may also have a potential role on infections [158]. Direct inhibitors of FXIa using small peptidomimetic molecules, monoclonal antibodies, aptamers, or natural inhibitors have been developed in recent years [159]. Preclinical data and rationale exist for preventing the activation of FXI and FXII preserving the hemostatic activity of FXI in COVID-19, and several inhibitors of FXII and FXI are currently under investigation [158] representing a promising therapeutic target against COVID-19 patients with severe disease.

As soon as the data from the RCTs are available, the therapy and prophylaxis recommendations will certainly be adapted and reissued.

## 8. Closing Remarks

COVID-19 can be considered a systemic disease characterized by the dysregulation of the immune system and a hypercoagulable status, a consequence of direct virus-induced endothelial damage, amplified by the leukocyte- and cytokine-mediated activation of the platelets, the release of TF, and NETosis and intensified by the activation of the complement system. The strong activation of the immune system by the SARS-CoV-2 infection leads to a non-regulatable thrombosis, which can present with many microthrombi in micro-vascularization, VTE, and arterial events. Coagulopathy is a crucial aspect of the disease, and its early identification, prevention, and treatment may limit its evolution towards potentially irreversible pulmonary and systemic conditions. Scientific evidence suggests that coagulopathy is not to be considered only as a disease complication but may be a real primitive pathogenetic element of SARS-CoV-2 infection. An important issue still to be addressed is long COVID, which is a common condition in patients who have been infected with SARS-CoV-2, regardless of the severity of the acute illness. A recent systematic review with metanalysis [160] found that most symptoms such as neurological symptoms, respiratory conditions, mobility impairment disorders with decreased exercise tolerance, heart conditions (palpitations), and general signs and symptoms, i.e., fatigue may be present with different frequencies, and the incidence is higher in females and increases with age. Among significant abnormalities identified through biochemical laboratory testing are increased levels of ferritin, C-reactive protein, and D-dimer [161]. Moreover, persistent dysfunctions of the immune response, with the chronic activation of T and B lymphocytes [162] and the presence of long-term immune system perturbations and autoimmunity [163] have been observed. The chronic low pro-inflammatory status has been related to endothelial and vascular alterations with a cytotoxic immune response towards endothelium [164]. Endothelium activation represents a significant risk of developing cardiovascular diseases for several months following infection. Recently, it was suggested the need to approach long COVID with non-pharmacological treatments, such as promoting physical activity [165]. The current knowledge of long COVID-19, though, does not allow stratifying patients into clusters that surely will benefit from exercise or have significant side effects. A better investigation of biomarkers modulated by exercise in long COVID-19 patients could be helpful to this end. Recent data from the literature also seem to suggest a favorable prognostic effect of anticoagulant treatment with low-molecular-weight heparin in patients with COVID-19 manifestations. The latter aspect is particularly pertinent in patients with cardiovascular and/or neurological diseases, obesity, or diabetes because they have a higher risk of developing vascular thrombosis. In conclusion, however, we underline that available data concerning anticoagulant treatment in COVID-19 are not completely supported by several randomized trials, and, therefore, there is an objective difficulty in choosing the most indicated therapy, which justifies a real advantage of a full-dose anticoagulant treatment in patients with severe disease, considering the potential risk of bleeding increase.

## Figures and Tables

**Figure 1 ijms-24-08945-f001:**
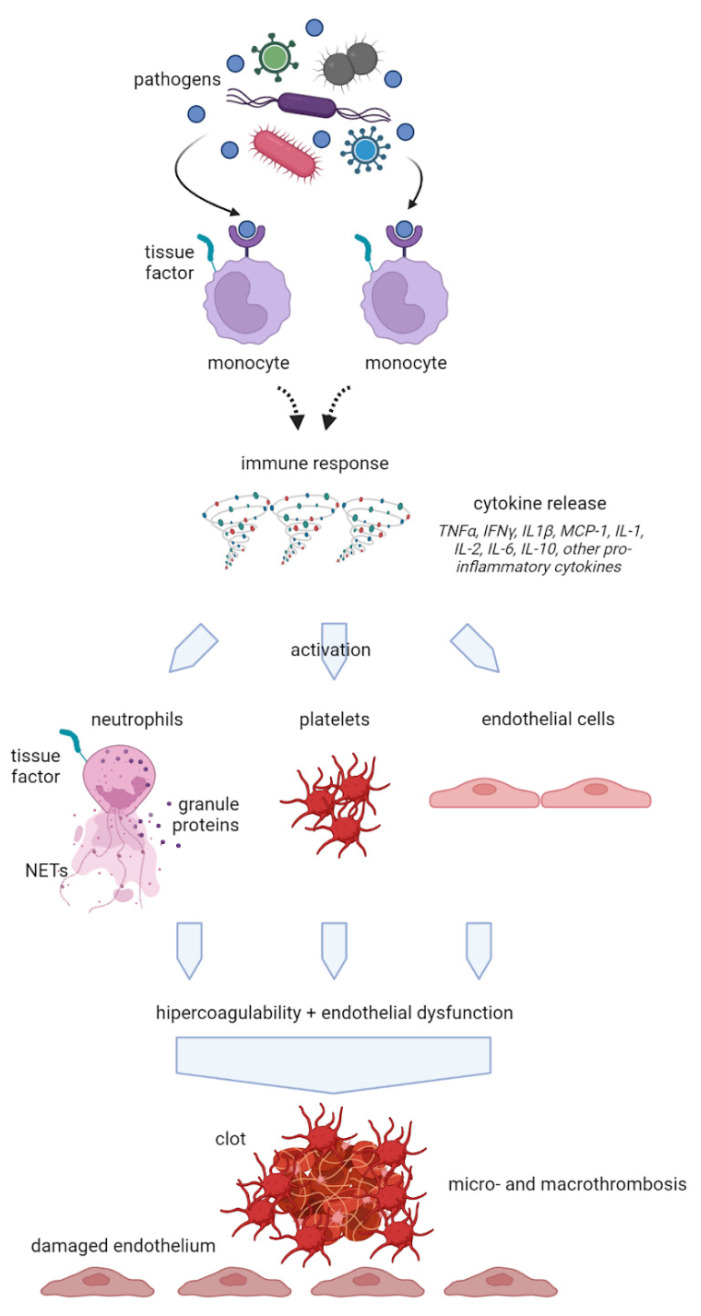
Schematic representation of endothelium activation towards a pro-thrombotic status. Pathogens and inflammatory mediators from injured host tissue activate monocytes and induce the expression of tissue factors on monocytes and endothelial cell surfaces. Subsequently, activated monocytes release inflammatory cytokines and chemokines amplifying the inflammatory response and stimulating vascular endothelial cells changing their properties to a procoagulant state. NETs: neutrophil extracellular traps; TNF*α*: tumor necrosis factor-alpha; IFNγ: interferon-gamma; MCP-1: monocyte chemotactic protein-1; IL: interleukin.

**Figure 2 ijms-24-08945-f002:**
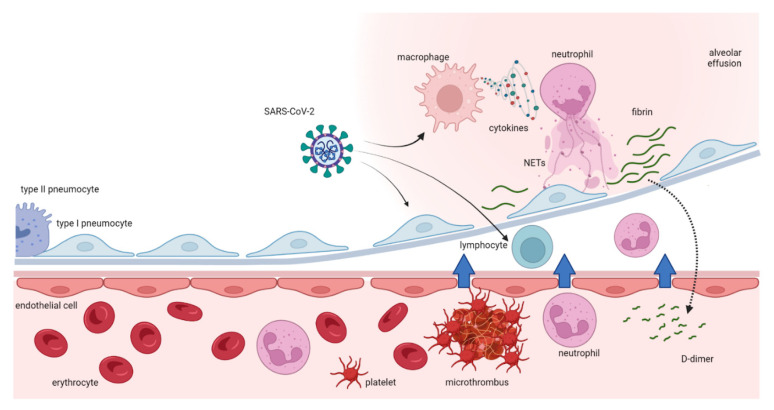
Schematic representation of the interlink between inflammatory and thrombotic mechanisms after COVID-19 infection.

**Figure 3 ijms-24-08945-f003:**
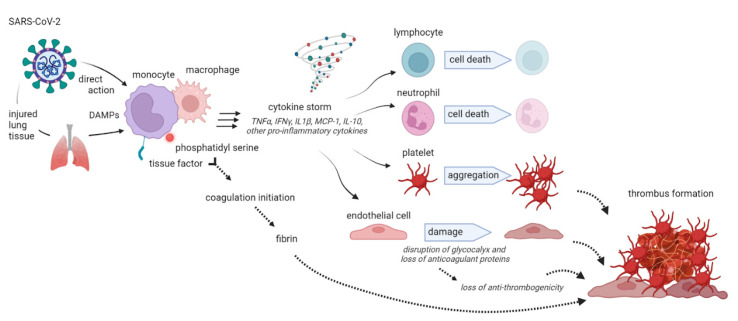
Effects of the inflammatory response to COVID-19 infection with cytokine release syndrome and dysregulation of the immune system with the final effect of a hyper-coagulative state. DAMPs: damage-associated molecular patterns.

**Figure 4 ijms-24-08945-f004:**
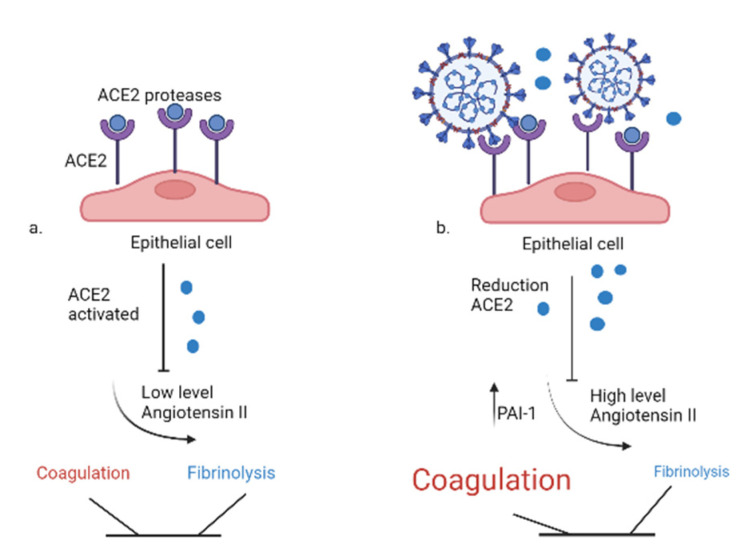
Imbalance between coagulation and fibrinolysis: the effects of SARS-CoV2. (**a**). Physiologically, ACE-2 reduces the availability of angiotensin II with no effects on coagulation and fibrinolysis. (**b**). SARS-CoV2 reducing ACE-2 availability, which increases the level of angiotensin II and the PAI-1 and favors the activation of the coagulation system. ACE-2: angiotensin-converting enzyme 2, PAI-1: plasminogen activator inhibitor 1.

**Table 1 ijms-24-08945-t001:** Sepsis-induced coagulopathy (SIC) score. ISTH score.

Item	Value	Score
SOFA score	1 ≥2	1 2 ≥4
PT-INR	1.2–1.4 >1.4	1 2
Platelet count (×mm^3^)	100,000–150,000 <100,000	1 2

INR: international normalized ratio; PT: prothrombin time; SOFA: sequential organ failure assessment.

**Table 2 ijms-24-08945-t002:** Increasing coagulation and inflammatory biomarkers.

**Coagulation biomarkers**	D-dimer, PLT, PT, APTT, FIB
**Inflammatory biomarkers**	ESR, CRP, Serum ferritin, PCT, IL-2, IL-6, IL8, IL10

Platelets (PLT), prothrombin time (PT), activated partial thromboplastin time (APTT), fibrinogen (FIB), erythrocyte sedimentation rate (ESR), C-reactive protein (CRP), procalcitonin (PCT), and interleukin (IL).

**Table 3 ijms-24-08945-t003:** Incidence of thrombotic events in patients with SARS-CoV-2 infections.

Study	Sample Size	Thrombotic Event Reported	Confirmatory Diagnostic Test	Incidence
Klok et al. [8]	N = 184 ICU patients	Venous arterial thrombosis	CTPA or Ultrasound	31%
Leonard-Lorant et al. [126]	N = 106 (48 ICU and 58 non-ICU)	Acute PE	CTPA	30% of all COVID-19 patients developed PE irrespective of ICU status
Helms et al. [9]	N = 150 ICU patients	Clinically significant thrombosis	CTPA	43%
Wichmann et al. [127]	N = 12 (5 ICU and 7 non-ICU)	DVT	Autopsy	58% of all COVID-19 patients autopsied had evidence of PE, irrespective of ICU status
	N = 156 non-ICU patients	DVT	Ultrasound	15%
Nahum et al. [128]	N = 34 ICU patients	DVT	Ultrasound	79%
Middeldorp et al. [129]	N = 198 (123 non-ICU and 75 ICU)	VTE in non-ICU vs. ICU	Ultrasound	9.2% in non-ICU vs. 59% in ICU
Shah et al. [130]	N = 187 (182 non-ICU and 5 ICU)	Acute PE	CTPA	23%
Cui et al. [131]	N = 81 non-ICU	DVT	Ultrasound	25%

ICU, intensive care unit; CTPA, computed tomography pulmonary angiogram; DVT, deep vein thrombosis; PE, pulmonary embolism; VTE, venous thromboembolism.

## Data Availability

No new data were created.

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
