# Peer review of "Interrelationship between COVID-19 and Coagulopathy: Pathophysiological and Clinical Evidence"

_ijms, 2023, doi:10.3390/ijms24108945_

Round 1

Reviewer 1 Report

In my opinion, the authors wrote an interesting and well-structured paper. I also appreciate a lot the topic that is relevant to the SARS CoV-2 infection. Also, figure 2 well describes the relationship between COVID-19 and coagulopathy.

Below are my suggestions:

Describe better some experiences that show the role of heparin in COVID-19 outcome (see and cite if you want). Heparin in COVID-19 Patients Is Associated with Reduced In-Hospital Mortality: The Multicenter Italian CORIST Study. Thromb Haemost. 2021 Aug;121(8):1054-1065. doi: 10.1055/a-1347-6070 and SARS-CoV-2 Transmission and Outcome in Neuro-Rehabilitation Patients Hospitalised at a Neuroscience Hospital in Italy. Mediterr J Hematol Infect Dis. 2020 Sep 1;12(1):e2020063. doi: 10.4084/MJHID.2020.063.

Than, the review is really well structured, and I suggest also adding information and explaining better the role of ACE-2 and angiotensin II during COVID-19 

Furthermore, the tables are interesting. Good representation of the topic. If you can add also information in paragraph 6 about the role of polymorphism and the COVID-19 outcome,

- Always use COVID-19 rather than COVID.

in closing remarks I believe that a short passage on long COVID is needed because of the high number of people with COVID-19 (see Incidence of long COVID-19 in people with previous SARS-Cov2 infection: a systematic review and meta-analysis of 120,970 patients). Intern Emerg Med. 2022 Nov 30:1–9. doi: 10.1007/s11739-022-03164-w.) and a passage also on the lack of therapy and the possible pathologic role of high D-dimer and interleukin on the onset of long COVID.

In conclusion, the paper after minor revision is suitable for publication in your relevant journal.

Minor English revision is needed

Author Response

Below are my suggestions:

Describe better some experiences that show the role of heparin in COVID-19 outcome (see and cite if you want). Heparin in COVID-19 Patients Is Associated with Reduced In-Hospital Mortality: The Multicenter Italian CORIST Study. Thromb Haemost. 2021 Aug;121(8):1054-1065. doi: 10.1055/a-1347-6070 and SARS-CoV-2 Transmission and Outcome in Neuro-Rehabilitation Patients Hospitalised at a Neuroscience Hospital in Italy. Mediterr J Hematol Infect Dis. 2020 Sep 1;12(1):e2020063. doi: 10.4084/MJHID.2020.063.

  • Thank you for the interesting observation. In accordance with the reviewer, we described with more details the role of heparin in COVID-19 outcome also adding references as proposed, in section 7 “New clinical evidences of anticoagulant therapy in COVID-19” (lines 483-495).

Then, the review is really well structured, and I suggest also adding information and explaining better the role of ACE-2 and angiotensin II during COVID-19 

  • Thanks to the reviewer, in accordance with the suggestion and for completeness we integrated the section 5.1.2. “Virus-specific mechanisms” providing information regarding the role of ACE-2 and angiotensin II in COVID-19 infection (lines 294-318).

Furthermore, the tables are interesting. Good representation of the topic. If you can add also information in paragraph 6 about the role of polymorphism and the COVID-19 outcome,

  • We thank the reviewer for the interesting observation. Section 6.1. “Coagulation biomarkers in SARS-CoV-2 infection: a predictive method”, as suggested and in accordance with reviewer, has been expanded explaining, more extensively, the role of different genetic polymorphisms in COVID-19 outcome and adding recent data and new references on the topic (lines 425-440).

- Always use COVID-19 rather than COVID.

  • Thank you for the comment. We agree with the reviewer and used always COVID-19 in accordance to the suggestion.

In closing remarks I believe that a short passage on long COVID is needed because of the high number of people with COVID-19 (see Incidence of long COVID-19 in people with previous SARS-Cov2 infection: a systematic review and meta-analysis of 120,970 patients). Intern Emerg Med. 2022 Nov 30:1–9. doi: 10.1007/s11739-022-03164-w.) and a passage also on the lack of therapy and the possible pathologic role of high D-dimer and interleukin on the onset of long COVID.

  • Thank you to the reviewer for this appropriate observation. These issues are now commented in Section 8. “Closing Remarks” with new references on the text (lines 563-582).

In conclusion, the paper after minor revision is suitable for publication in your relevant journal.

Reviewer 2 Report

The article is of an overview nature and is devoted to an important aspect of practical healthcare - coagulopathy in covid infection. The authors have well selected the most significant scientific articles on this topic and conducted a very good analysis. It is advisable to provide information on the role of inflamosomes in the development of the inflammatory process in covid-19 coagulopathy, the possibility of using direct blockers of 11a factors.

Author Response

The article is of an overview nature and is devoted to an important aspect of practical healthcare - coagulopathy in covid infection. The authors have well selected the most significant scientific articles on this topic and conducted a very good analysis. It is advisable to provide information on the role of inflamosomes in the development of the inflammatory process in covid-19 coagulopathy, the possibility of using direct blockers of 11a factors.

  • Thank you to the Reviewer for the time spent to review our manuscript and the precious comments to improve it. In accordance with the Reviewer suggestion, we expanded the section 7 “New clinical evidences of anticoagulant therapy in COVID-19” (lines 528-548) adding information and new references about the role of the contact activation system linking inflammation and coagulopathy in COVID-19 infection and the promising role of direct FXI inhibitors in controlling COVID-19 severe inflammatory and thrombotic consequences.
